# Arterial Thrombotic Complications in COVID-19: A Case of Renal Infarction

**DOI:** 10.3390/biomedicines10102354

**Published:** 2022-09-21

**Authors:** Mariangela Mancini, Gianmarco Randazzo, Gregory Piazza, Fabrizio Dal Moro

**Affiliations:** 1Urological Clinic, University Hospital of Padova, 35121 Padova, Italy; 2Department of Surgical, Oncological and Gastroenterological Sciences, University of Padova, 35121 Padova, Italy; 3Division of Cardiovascular Medicine, Brigham and Women’s Hospital, Harvard Medical School, Boston, MA 02115, USA

**Keywords:** COVID-19, renal infarction, rare urology, thromboembolic complications, platelets activation

## Abstract

COVID-19 infection has been associated with thrombotic complications, especially venous thromboembolism. Although arterial thrombotic complications are rarely seen in these patients, we report the case of a 43-year-old patient who developed thrombosis of the main branch of the left renal artery, causing partial infarction of the left kidney associated with severe pain. He had no risk factors for thrombosis except for COVID-19 infection. We excluded any possible condition usually associated with renal artery thrombosis/embolism (i.e., cardiovascular, oncological, hematological, or rheumatic). The thrombosis resolved after a combination of anticoagulant and anti-platelet therapy. This case highlights the importance of the risk of recurrence of thrombosis in patients with a recent history of COVID-19, even after hospital discharge, improvement of the initial thrombotic event, and clearance of SARS-CoV-2 infection.

## 1. Introduction

Spontaneous renal infarction is a rare condition, with an estimated incidence of 0.004–0.007% based on visits to emergency departments [1].

SARS-CoV-2 infection has been associated with increased frequency of both venous and arterial thrombotic complications, especially in critically ill hospitalized patients.

Systemic inflammation, immobility, and hypercoagulable states play an important role in thromboembolic complications in COVID-19 infection [2,3]. Frequently, patients with COVID-19 are noted to have increased D-dimer, prothrombin, and fibrinogen levels, which speak to activated coagulation [4].

The pathophysiology of thromboembolism in COVID-19 compared with other non-COVID-19 disorders may also be more associated with platelet activation and viral-mediated endothelial inflammation and injury [5].

Endotheliopathy is suspected to be due directly to viral infection, which is promoted by the overexpression of angiotensin-converting enzyme receptor 2, the receptor for cell entry of SARS-CoV-2 [6].

Moreover, systemic hyperinflammation produces a “cytokine storm syndrome” (tumor necrosis factor, interleukin [IL]-6, and IL-1β), which may contribute to the development of intravascular coagulopathies [7].

## 2. Case Report

### 2.1. First Hospital Admission

We report the case of a 43-year-old patient who was admitted to the Emergency Department with severe sudden-onset left flank pain, during a paucisymptomatic COVID-19 infection, which had been diagnosed 3 days before hospital admission. The diagnosis was confirmed upon hospital admission. He had had a fever up to 39 °C in the previous days and he had been vaccinated with three doses of Pfizer/BioNTech Comirnaty vaccine (BNT162b2) by the onset of symptoms.

A contrast-enhanced CT scan was performed in the Emergency Department, showing a 7 cm hypodense area at the mid-upper third of the left kidney consistent with renal infarction (Figure 1).

He had no previous medical history of thrombotic risk factors, and he did not smoke. He had moderate stenosis of a bicuspid aortic valve and a small non-functioning adrenal adenoma.

He was admitted to the Infective Diseases Unit for further diagnostic workup.

Clinical examination showed tenderness on pressure on the left flank and no clinical signs of deep vein thrombosis.

Initial blood tests revealed a leukocytosis of 12.27 × 10^9^/L, an elevated C-reactive protein of 127.6 (normal range of 0.00–5.00 mg/L), a normal procalcitonin, a creatinine of 75 µmol/L (eGFR of 99 mL/min/1.73 m^2^), a normal INR, and a normal aPTT (Table 1).

The coagulation panel showed increased levels of coagulation factors VIII, IX, and XI and fibrinogen activity.

The autoantibodies panel (ANA, ENA, ANCA, RF, C3, C4, anti-phospholipid antibodies, anti-beta2 glycoprotein-1, circulating immunocomplexes, homocysteine, LpA cryoglobulins) was normal.

Blood cultures showed no growth. Other tests for antistreptolysin O titer, Widal–Wright reaction, antibodies for Coxiella, Bartonella, and Legionella urinary antigen were all negative.

An electrocardiogram showed no signs of atrial fibrillation, and a trans-thoracic echocardiogram showed no source of embolism, moderate aortic stenosis, and bicuspid aortic valve. Broad-spectrum antibiotics were started (piperacillin-tazobactam, 4.5 g every 8 h. For analgesia, tramadol (200 mg) plus ketoprofen (400 mg) every 48 h were administered. Anticoagulant therapy with LMWH was started at a dose of 7000 UI b.i.d.

A contrast-enhanced CT scan performed five days later showed stability of the ischemic area on the left kidney, with a delayed nephrogram on the left but a symmetrical excretory phase by both kidneys. A filling defect in a segmental collateral of the left renal artery was reported.

After resolution of symptoms, the Patient was discharged 8 days after admission on LMWH therapy (7000 UI b.i.d.) and a prescription to undergo a repeat test for COVID-19 and a trans-esophageal echocardiogram.

A molecular test for COVID-19 was negative 2 days after discharge. He was asymptomatic at home.

### 2.2. Second Hospital Admission

One month later he was re-admitted to the ER for severe acute left flank pain.

A COVID-19 nasopharyngeal swab PCR test was negative on admission.

A contrast-enhanced CT scan was performed, showing proximal extension of the left renal artery thrombus (now occluding one of the two main branches of the artery) with an increase of the left renal parenchymal infarction (Figure 2).

The patient was admitted to the Urology, part of the ERN eUROGEN, a European network for the treatment of rare and complex urological conditions [8,9]. Blood levels of LMWH (measured at 4 h after subcutaneous self-administration) were found to be 579 U/L (reference range: 600–1000 when administered twice daily). The dose of Enoxaparin was therefore increased to 8000 UI twice daily (reaching a level of 695 U/L). Aspirin 100 mg daily was started. Broad-spectrum antibiotic prophylaxis with piperacillin-tazobactam (4.5 g every 6 h) was added.

A vascular surgery consultation did not recommend any vascular or endovascular procedure, due to the high risk of further thrombotic complications. To achieve satisfactory pain control, 50 mg of morphine every 48 h IV was added.

During hospitalization, additional causes of left renal artery thrombosis/embolization other than COVID-19 were explored.

A cardiothoracic surgery consultation and a subsequent trans-esophageal echocardiogram excluded the presence of intracardiac thrombi or endocarditis. Multiple sets of blood cultures were negative for growth.

To exclude a paraneoplastic syndrome, a PET-CT with FDG was performed, which did not show areas of focal/pathological uptake of the tracer. Oncological markers, such as AFP, CEA, CA 19-9, PSA, beta-HCG, aldosterone, and 24-h urinary metanephrines, were in the normal range (Table 2). Moreover, we performed sequential multimodal imaging studies during hospitalization (Figure 2, Figure 3 and Figure 4).

A contrast-enhanced ultrasound at admission showed diffused cortical ischemic areas max 1 cm in diameter at the middle third of the left kidney. A repeat ultrasound performed after 10 days showed improvement of left kidney perfusion, with a small capsular artery now supplying the ischemic area at the middle third. This was confirmed by a follow up-ultrasound performed 30 days after discharge.

The patient remained under pain control medications for 10 days. Gradually the pain resolved, and the medications were discontinued. LMWH was replaced with edoxaban (60 mg daily). Aspirin was maintained at the dose of 100 mg daily. Antibiotics were continued with amoxicillin/clavulanate 875/125 every 8 h for 10 days after discharge. The patient was discharged on the 21st day of hospitalization. At second discharge, creatinine levels and eGFR were normal (serum creatinine: 103 µmol/L; eGFR 78.6 mL/min/1.73 m^2^). After discharge, he recovered without further symptoms and resumed his work activity (university teaching) without complications.

After 30 days, a follow-up CT scan showed resolution of the renal artery thrombus with improved vascularization of the left kidney, and persistence of residual infarcted areas (Figure 2D–H). DMSA static renal scintigraphy showed a small left kidney with an estimated residual relative renal function of 28% (Figure 4).

## 3. Conclusions

In the absence of other possible causes which can explain renal artery thrombosis/embolism, the most probable cause of renal infarction in this case was COVID-19.

While venous thromboembolism is a well-documented complication of COVID-19 [3], arterial thrombosis is less frequently recognized as a complication [4]. Like venous thromboembolism, arterial thrombosis may occur even in non-hospitalized patients, as shown in this case report.

Arterial thrombosis may result following COVID-19 even in the absence of pro-thrombotic medical risk factors. Renal infarction should be considered in the differential diagnosis for patients presenting to the ER with acute flank pain and COVID-19.

Our patient started to improve after the adjustment of anticoagulation and the addition of antiplatelet therapy. This is consistent with the theory that platelet activation may also play a crucial role in the pathogenesis of thromboembolic complications in COVID-19 [2].

That said, antiplatelet therapy for prophylaxis of thromboembolism during COVID-19 is not recommended by current guidelines [10,11].

Our case seems to suggest that careful follow-up is needed for patients with COVID-19 and thrombosis, even after hospital discharge, and the clearance of SARS-CoV-2 infection. Recent data shows clearly that the inflammatory cascade and associated risk of thrombosis remain increased for many weeks after infection [12].

The possible additional advantage of antiplatelet therapy in this setting has been and continues to be considered in ongoing research studies. Further studies and reports may help to develop a better strategy for treating similar complications of COVID-19 and prevent recrudescence of symptoms or worsening of the initial thrombotic process.

## Figures and Tables

**Figure 1 biomedicines-10-02354-f001:**
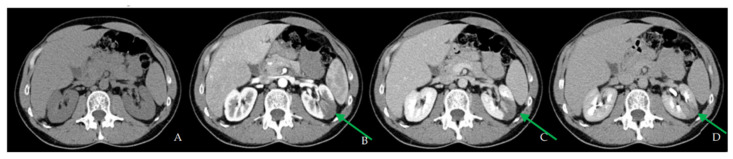
CT scan performed at first admission ((**A**) basal; (**B**) arterial phase; (**C**) venous phase; (**D**) urographic phase) shows an ischemic hypodense area in the upper-middle third of the left kidney (green arrows).

**Figure 2 biomedicines-10-02354-f002:**
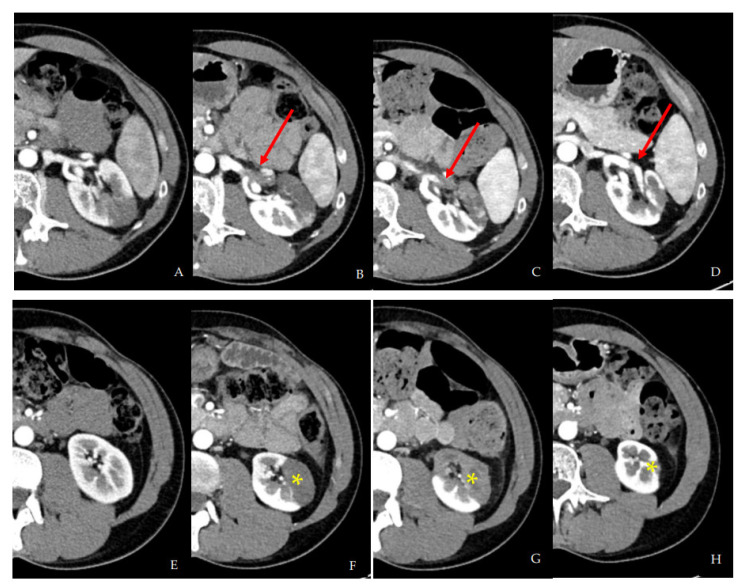
(**A**) shows first CT scan on first admission. (**B**–**D**) (red arrows) show thrombosis of one of the two branches of left renal artery at second hospital admission (**B**), after 7 days (**C**), and after 30 days, with almost complete recanalization (**D**). (**E**) shows lower third of renal parenchyma on first admission; (**F**–**H**) (yellow asterisks) shows evolution of left renal infarction involving the lower third of renal parenchyma ((**F**) second hospital admission; (**G**) 7 days; (**H**) 30 days).

**Figure 3 biomedicines-10-02354-f003:**
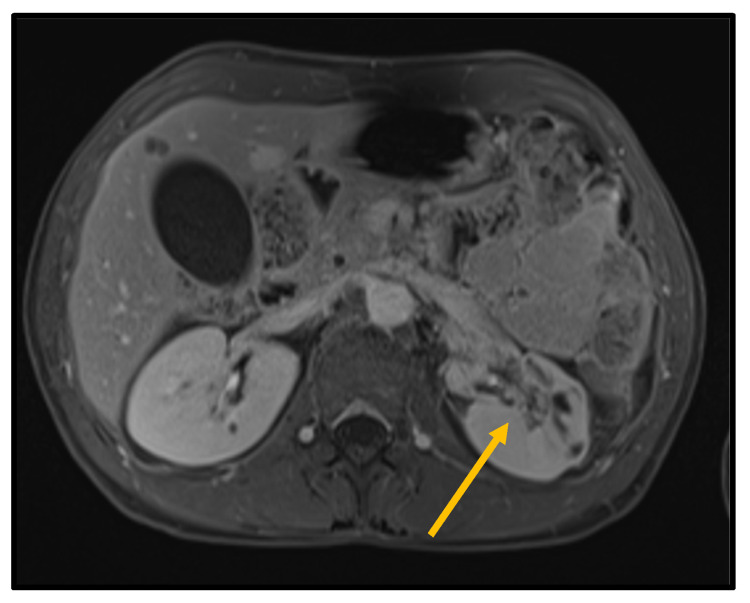
Contrast-enhanced MRI scan confirming anterolateral ischemic area involving especially renal medulla (orange arrow).

**Figure 4 biomedicines-10-02354-f004:**
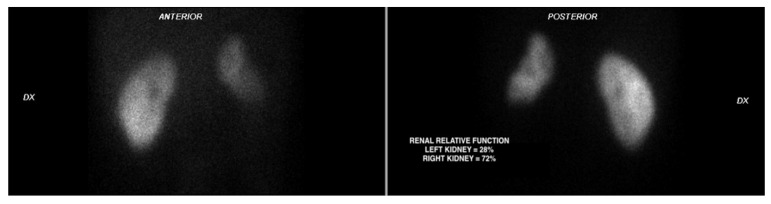
DMSA static renal scintigraphy at 30 days from discharge shows a small left kidney with reduced uptake of the tracer.

**Table 1 biomedicines-10-02354-t001:** An increase in inflammatory markers is compatible with renal infarction. Immunological markers were all negative. An increase in coagulation factors activity, especially factor II, VIII, IX, X, X, and fibrinogen is coherent with the pathophysiology of thromboembolism described in COVID-19 [8,9,10].

	Value	Reference Range
Inflammatory markers		
Leukocytes, ×10^9^/L	12.3	4.4–11
CRP, mg/dL	127.6	0–5
Procalcitonin, µg/L	0.08	<0.5
Creatinine, µmol/L	75	59–104
ESR, mm/h	46	2–28
LDH, U/L	462	135–225
Immunological markers		
S-ICC-C1q, µgEq/mL	1.2	<16
S-ICC-C3d, µgEq/mL	1.7	<16
C3 factor, g/L	1.6	0.9–1.8
C4 factor, g/L	0.32	0.09–0.36
Plasmatic crioglobulins	Negative	
Rheumatoid factor, kU/L	<10	0–30
Anti-ENA ab	Negative	
Anti-ANCA ab	Negative	
Anti-proteinase III ab	Negative	
Anti-MPO ab	Negative	
Anti-native DNA ab	Negative	
Coagulation panel		
INR	1.12	
aPTT, s	23	22–32
D-dimer, µg/L	<150	0–250
Factor II activity, %	134.8	80–120
Factor VIII activity, %	265.3	60–160
Factor IX activity, %	175.4	80–120
Factor X activity, %	117.6	80–120
Factor XI activity, %	153.4	80–120
Fibrinogen (Clauss assay), mg/dL	681.2	150–450
Antitrombin activity, %	101.2	80–120
Protein C, %	109	80–120
Protein S, %	118	80–120
Factor V Leiden mutation	Negative	
Protrombin variant G20210A	Negative	
Plasminogen activity, %	120.4	75.0–140
Anti beta-2 GPI IgG, U/mL	8.6	<8
Anti beta-2 GPI IgM, U/mL	1.7	<8
Anti cardiolipin IgG, U/mL	8.2	<10
Anti cardiolipin IgM, U/mL	2.1	<10

Abbreviations: CRP, C-reactive protein; ESR, Erythrocyte Sedimentation Rate; LDH, Lactate Dehydrogenase.

**Table 2 biomedicines-10-02354-t002:** Oncological markers.

	Value	Reference
Oncological Markers		
AFP, μg/L	3.0	0.0–7.4
CA-19-9, kU/L	12.5	0.0–30.0
CEA, μg/L	1.1	0.0–4.0
24-h urine metanephrines μmol/L	0.48	0.01–1.62
beta-HCG, IU/L	<0.05	0.0–1.5
PSA, μg/L	0.41	0.02–4.000

Abbreviations: AFP, alphafetoprotein; CA-19-9, cancer antigen 19-9; CEA, Carcino-Embrionic Antigen; HCG, human chorionic gonadotropin; PSA, prostate specific antigen.

## Data Availability

Available at Padova Hospital records upon request.

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
