# Peer review of "Arterial Thrombotic Complications in COVID-19: A Case of Renal Infarction"

_biomedicines, 2022, doi:10.3390/biomedicines10102354_

Round 1
Reviewer 1 Report
In this case report, Mancini et al. reported a COVID-19-infected patient with repeated renal infarction. This report provided a considerable clinical significance for the attention of recurrence of thrombosis in patients with a history with COVID-19 infection once previous thrombosis were improved.
Comments
1) There is no indication of lower-cases in Fig.1 and 2. Please indicate them.
2) In figures, please clarify the parts where the authors would like to indicate by adding arrows, an so on.
3) How about creatinine levels (and eGFR) at 2nd discharge ?
Reviewer 2 Report
I want to commend the authors for their concise case presentation. I have only one observation. It is unclear to me when the patient was diagnosed with COVID-19 since it is not apparent from the text when this positive result was available to make the diagnosis during the initial hospitalization. Also upon re-hospitalization for the second time, how is this explained in absence of a positive test since as mentioned for this hospitalization this was negative. I would also recommend authors for consistence to use caps when referring to the virus, in introduction both Covid and COVID are used and this should be corrected so only the latter is used through the text.
